# Management of children visiting the emergency department during out-of-office hours: an observational study

Gina Schinkelshoek [1] , Dorine M Borensztajn [1] , Joany M Zachariasse [1] ,
Ian K Maconochie,[2] Claudio F Alves,[3] Paulo Freitas,[4] Frank J Smit,[5]
Johan van der Lei,[6] Ewout W Steyerberg,[7] Susanne Greber-Platzer,[8]
Henriëtte A Moll [1]

► Additional material is published online only. To view please visit the journal online (http://dx.doi.org/10.1136/bmjpo-2020-000687).

For numbered affiliations see end of article.

**Correspondence to**
Prof. Dr. Henriëtte A Moll; h.a.moll@erasmusmc.nl

## ABSTRACT

**Background** The aim was to study the characteristics and management of children visiting the emergency department (ED) during out-of-office hours.

**Methods** We analysed electronic health record data from 119 204 children visiting one of five EDs in four European countries. Patient characteristics and management (diagnostic tests, treatment, hospital admission and paediatric intensive care unit admission) were compared between children visiting during office hours and evening shifts, night shifts and weekend day shifts. Analyses were corrected for age, gender, Manchester Triage System urgency, abnormal vital signs, presenting problems and hospital.

**Results** Patients presenting at night were younger (median (IQR) age: 3.7 (1.4–8.2) years vs 4.8 (1.8–9.9)), more often classified as high urgent (16.3% vs 9.9%) and more often had ≥2 abnormal vital signs (22.8% vs 18.1%) compared with office hours. After correcting for disease severity, laboratory and radiological tests were less likely to be requested (adjusted OR (aOR): 0.82, 95% CI 0.78–0.86 and aOR: 0.64, 95% CI 0.60–0.67, respectively); treatment was more likely to be undertaken (aOR: 1.56, 95% CI 1.49–1.63) and patients were more likely to be admitted to the hospital (aOR: 1.32, 95% CI 1.24–1.41) at night. Patterns in management during out-of-office hours were comparable between the different hospitals, with variability remaining.

**Conclusions** Children visiting during the night are relatively more seriously ill, highlighting the need to keep improving emergency care on a 24-hour-a-day basis. Further research is needed to explain the differences in management during the night and how these differences affect patient outcomes.

## INTRODUCTION

Concerns exist that emergency care may differ during out-of-office hours, that is, evening, night and/or weekend shifts compared with office hours. For example, higher mortality rates were found during weekend visits in adult patients with sepsis presenting to the emergency department (ED).[1] Some suggest that worse patient outcomes during out-of-office hours can, partly, be explained by

### What is known about the subject?

► Concerns exist that emergency care may differ during out-of-office hours and office hours. There has been growing attention for paediatric emergency care at the emergency department (ED) recently. However, little is known about children visiting the ED during out-of-office hours.

### What this study adds?

► Children visiting the ED during the night were relatively more seriously ill than during office hours. Fewer diagnostic tests, more treatment and more hospital admissions were reported during the night, which could not be explained by correcting for illness severity.

higher illness severity and not by the time of arrival as an independent factor.[2 3] When looking at the paediatric population, higher mortality rates were observed among children admitted to the paediatric intensive care unit (PICU) during out-of-office hours.[4] However, other studies analysing PICU patients are in conflict with this finding.[5–9]

Over the past few years, increasing attention has been paid to paediatric emergency care, leading to different guidelines and standards for children visiting the ED.[10–14] These guidelines aim to achieve optimal utilisation of available resources, optimise patient outcomes and thereby improve the quality of care. A recent study reported that main standards of care are fulfilled in European EDs, but some areas of improvement still exist.[15]

However, little research has focused specifically on children visiting the ED during out-of-office hours. Recently, variation in paediatric ED utilisation patterns and the presentation of these children by the shift of arrival has

been reported.[16] But this study was a single-centre study, which may limit generalisability. Moreover, no study has analysed resource utilisation at paediatric EDs during out-of-office hours, except for studies analysing children with a specific presenting problem.[17–19]

It is important to obtain insight into the characteristics and management of the whole paediatric patient population attending the ED with its variable presenting problems, as this may be helpful in optimising guidelines and resource allocation, thereby improving the quality of care on a 24-hour-a-day basis.

The aim of this study was to compare patient characteristics and management of paediatric patients visiting the ED during out-of-office hours to office hours in four different European countries.

## METHODS
### Study design, setting and patients
This study is part of the TrIAGE project (Triage Improvement Across General Emergency departments for paediatric patients), a prospective observational study.[20] This project captured routinely collected, standardised, electronic health record data of all non-scheduled ED visits of children <16 years in five different hospitals in four different countries. The five participating hospitals, selected by convenience sampling, were: Erasmus Medical Centre, the Netherlands; Maasstad Hospital, the Netherlands; St Mary's Hospital Imperial College Healthcare National Health Service Trust, UK; Hospital Professor Doutor Fernando Fonseca, Portugal; and Vienna General Hospital, Austria. The enrolment period varied from 8 to 36 months between 2012 and 2015.

At the beginning of the study, a minimum dataset of variables for the collection was defined. If necessary, entries in the electronic health data were added or modified. Nurses at the ED were informed about the study and encouraged to fill in the entire medical records, including vital signs, which is in principle routine data. During the study, these routinely collected data were documented in the electronic, medical records by nurses at the ED. Data completeness and accuracy of the data were reviewed using a checklist of quality control. Sometimes, data were completed by information or vital signs from medical records documented by physicians at the ED. Coded data were extracted from the electronic hospital information systems and thereafter transferred to the principal investigator as an encrypted file. Careful data harmonisation and quality checks were conducted. The requirement for informed consent was obtained by all hospitals and the study was approved by the medical ethical committees of all participating institutions.

### Patient and public involvement statement
Patients or the public were not involved in this research.

### Definitions
#### Office hours versus out-of-office hours
Office hours were defined as Monday until Friday, between 08:00 and 17:59, also referred to as day shift week. We compared office hours, as reference shift, to out-of-office hours, divided into evening shifts, night shifts and day shifts during the weekend. Evening shifts were defined as Monday until Sunday between 18:00 and 22:59 and night shifts as Monday until Sunday between 23:00 and 07:59, based on the previous literature.[5 9 17] Weekend day shifts were defined as Saturday and Sunday between 08:00 and 17:59.

#### Patient characteristics
For this study, the following data were extracted from the database: age, gender, time and date of arrival, Manchester Triage System (MTS) urgency, vital signs and presenting symptom (based on the MTS flowchart). The MTS urgency consists of five categories: emergent, very urgent, urgent, standard and non-urgent.[21–23] These categories were divided into three groups: high urgent (emergent or very urgent, <10 min waiting time), urgent (urgent, <60 min waiting time) and low urgent (standard or non-urgent, ≥60 min waiting time). Vital signs included heart rate, respiratory rate, oxygen saturation and temperature. Vital signs were considered abnormal based on the Advanced Paediatric Life Support reference values, with fever defined as temperature ≥38.5°C.[24] The MTS flowchart was used to create subgroups of presenting problems. These subgroups, based on a previous study,[25] included medical (respiratory, ear, nose and throat problems, gastrointestinal problems, neurological or psychiatric problems, general malaise, urological or gynaecological problems, dermatological and others) and trauma (trauma or muscular and wounds) (online supplementary table 1.1).

#### Management
Management was defined as diagnostics (laboratory tests and imaging), treatment at the ED and hospital or PICU admission directly after ED visit. Laboratory tests were divided into blood tests, including cultures, and urine tests, including cultures. Imaging was divided into: X-ray, ultrasound and CT.

Treatment was divided into: oral medication, inhalation medication, intravenous medication or intravenous fluids and immediate lifesaving interventions (ILI) (online supplementary table 2).[26]

In the multivariable regression models, we defined laboratory diagnostics as any form of laboratory tests, radiological diagnostics as any form of imaging and treatment as any form of treatment.

Hospital admission included patients who were admitted to the general ward or the PICU. PICU admission was defined as admission to the PICU directly from the ED.

### Statistical analysis
In descriptive statistics of patient characteristics, continuous, skewed variables are shown as median with IQR.

**Table 1** Missing data for vital signs and imputation model

| Vital sign | Missing % of total patients (interhospital range) |
|---|---|
| Heartrate | 42.7 (19.5–62.9) |
| Respiratory rate | 52.4 (23.5–87.4) |
| Oxygen saturation | 44.9 (19.7–66.4) |
| Temperature | 22.5 (5.9–58.2) |

Missing data for vital signs were imputed using a multiple imputation model including patient characteristics, date and time, triage items, vital signs, diagnostics, therapy and disposition after the ED. This imputation process resulted in 25 datasets on which statistical analyses were performed and pooled for a final result.[27] Imputation was performed by using the MICE imputation package in R V.2.15.2.

ED, emergency department.

Nominal and ordinal variables are shown as the number of patients with percentage within the shift. Nominal and ordinal variables considering management are shown as percentages within the shift, with interhospital ranges.

In multivariable regression analyses, differences in management were compared between office hours and evening, night and weekend day shifts. To study the independent role of the shift of arrival, the regression analyses were adjusted for age, gender, MTS urgency, abnormal vital signs, presenting problem and hospital. We performed stratified analyses, by conducting the adjusted regression analyses for each hospital separately, to provide insight into interhospital differences.

Missing data for vital signs were imputed using a multiple imputation model (table 1). These missing data were assumed to be conditional on other variables included in the database, that is, missing at random (online supplementary 3). Sensitivity analyses were performed by analysing a database restricted to complete cases (online supplementary 4).[27 28]

Results were considered statistically significant at a p value of ≤0.05.

Initial analyses were performed with SPSS software V.21 (IBM SPSS Statistics, IBM Corporation).

## RESULTS

### Patient population

In total, 119 209 patients, ≤16 years, visited the participating EDs. Five patients were excluded owing to missing arrival time or date, leaving 119 204 patients for analysis, with 50 417 (42.3%) of these patients who visited the ED during office hours versus 68 787 (57.7%) during out-of-office hours. Of those, 18 220 (15.3%) visited during weekend day shifts, 36 429 (30.6%) during the evening and 14 138 (11.9%) during the night.

Patients who visited the ED during the night were younger, were more often classified by MTS as high urgent and more often had two or more abnormal vital signs (table 2).

During office hours, general malaise (20.0%), trauma or muscular (19.4%) and gastrointestinal problems (15.1%) were the most common presenting problems (figure 1). During the night, relatively more general malaise (24.6%), gastrointestinal problems (20.2%) and respiratory problems (17.6%) and fewer trauma cases (8.2%) were observed. Presenting problems during the evening and weekend day shifts were similar compared with office hours (online supplementary table 1.2).

### Management

During out-of-office hours, fewer diagnostic tests, more treatment and more hospital and PICU admissions were conducted (table 3).

**Table 2** Patient characteristics

| Total (n=119 204) | Office hours* (n=50 417) | Evening shift (n=36 429) | Night shift (n=14 138) | Day shift (n=18 220) |
|---|---|---|---|---|
| Median age, years (IQR) | 4.8 (1.8–9.9) | 4.5 (1.8–9.5) | 3.7 (1.4–8.2) | 4.2 (1.7–8.8) |
| Gender, n (%) | | | | |
| Male | 27 150 (53.9) | 19 698 (54.1) | 7566 (53.5) | 9956 (54.6) |
| MTS, n (%) | | | | |
| High urgent | 4963 (9.9) | 4136 (10.7) | 2302 (16.3) | 1683 (9.2) |
| Urgent | 13 300 (26.4) | 10 796 (29.6) | 3655 (25.9) | 4866 (26.7) |
| Low urgent | 31 478 (61.7) | 21 276 (58.4) | 7892 (55.8) | 11 366 (62.4) |
| Missing | 676 (1.3) | 452 (1.2) | 289 (2.0) | 305 (1.7) |
| Vital signs, n (%) | | | | |
| Normal | 18 734 (37.2) | 12 903 (35.4) | 4812 (34.0) | 6553 (36.0) |
| 1 Abnormal | 22 545 (44.7) | 15 947 (43.8) | 6100 (43.1) | 8119 (44.6) |
| 2 or more abnormal | 9139 (18.1) | 7578 (20.8) | 3227 (22.8) | 3548 (19.5) |

*Office hours=day shift week.
MTS, Manchester Triage System.

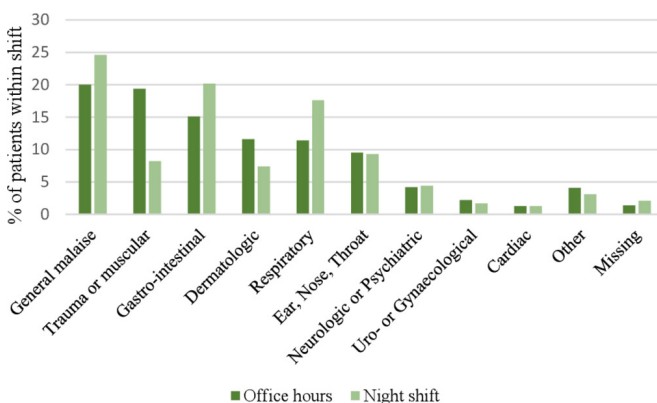

**Figure 1** Presenting problems.

During the evening and weekend day shift, fewer urine and blood tests were performed. During the evening, night and weekend day shifts, imaging was performed less often, except for X-ray during the evening shift. Treatments with oral, inhaled and intravenous medication were given more often, especially during the night. Also, during the night, slightly more ILIs were seen, while more hospital and ICU admissions were observed.

In multivariable regression analyses, the results were adjusted for age, gender, urgency, abnormal vital signs, presenting problem and hospital (table 4).

Laboratory and imaging diagnostics were significantly less likely to be performed during the evening and night shifts. Also, patients were significantly more likely to receive treatment during all out-of-office hours shifts and more likely to be admitted to the hospital during the night shift. After adjusting, no association was found between PICU admission and out-of-office hours visits.

When analysing a database consisting only of complete cases for vital signs, overall results were comparable with the analyses in the imputation database (online supplementary 4).

### Interhospital analyses

Hospital characteristics are shown in table 5 and online supplementary table 5.1-5.5. Some interhospital variation in management was observed (table 2). The multivariable regression analyses, adjusted for age, gender, urgency level, abnormal vital signs and presenting problem, were performed for each hospital separately in stratified analyses (online supplementary table 6). Overall patterns in management during out-of-office hours were comparable between the different hospitals, with some variation remaining. For example, at two hospitals, no significant difference was found in hospital admissions during the night. At another hospital, imaging was requested more frequently during evening and weekend shifts.

**Table 3** Management (diagnostics, treatment, hospital admission and PICU admission), descriptive

| Total (n=119 204) | Office hours* (n=50 417) | Evening shift (n=36 429) | Night shift (n=14 138) | Day shift weekend (n=18 220) |
|---|---|---|---|---|
| **Mean % (interhospital range)** | | | | |
| *Laboratory diagnostics* | | | | |
| Any laboratory test (n=119 204) | 22.2 (11.7–40.6) | 18.6 (9.2–33.0) | 21.5 (9.8–40.0) | 20.2 (8.0–33.3) |
| Urine tests (n=50 473)† | 20.2 (11.0–23.5) | 17.1 (8.1–22.3) | 20.2 (13.9–23.2) | 16.6 (7.7–19.6) |
| Blood tests (n=66 029)† | 21.9 (11.7–26.7) | 15.2 (9.2–20.3) | 20.4 (9.8–30.5) | 17.2 (7.9–23.5) |
| *Imaging diagnostics* | | | | |
| X-ray (n=119 204) | 19.5 (6.0–24.7) | 19.4 (3.7–40.9) | 12.1 (4.3–17.1) | 17.4 (5.5–41.2) |
| Ultrasound (n=119 204) | 2.7 (1.5–6.0) | 1.7 (0.5–2.9) | 1.2 (0.5–4.1) | 2.2 (0.6–4.2) |
| CT (n=119 204) | 1.2 (0.7–2.5) | 1 (0.6–2.4) | 0.9 (0.4–2.3) | 1 (0.5–2.1) |
| *Treatment* | | | | |
| Oral medication (n=108 621)† | 13.9 (8.0–34.3) | 17.2 (8.6–41.8) | 23.5 (14.9–45.1) | 16.5 (8.2–39.7) |
| Inhaled medication (n=119 204) | 6.5 (3.6–8.7) | 6.5 (3.5–9.0) | 12 (6.0–15.4) | 6.9 (3.5–9.8) |
| IV medication or fluids (n=119 204) | 7.3 (4.1–14.0) | 7 (4.2–11.3) | 11.3 (4.6–19.4) | 7.8 (3.4–12.0) |
| ILI (n=119 204) | 0.3 (0.0–1.1) | 0.3 (0.0–1.2) | 0.5 (0.0–1.9) | 0.4 (0.1–1.2) |
| *Hospital admission (n=119 204)* | | | | |
| Yes | 9.7 (4.3–22.6) | 10.2 (5.0–24.2) | 14.3 (8.8–38.4) | 9.9 (4.6–22.4) |
| *PICU admission (n=119 204)* | | | | |
| Yes | 0.6 (0.1–2.3) | 0.5 (0.1–2.8) | 0.8 (0.2–4.7) | 0.6 (0.0–2.7) |

*Office hours=day shift week.
†Not all variables were complete for all hospitals, making the total number less than 119 204.
ILI, immediate lifesaving interventions; IV, intravenous; PICU, paediatric intensive care unit.

**Table 4** The association between management (laboratory tests, imaging, treatment, hospital and PICU admission) and time of arrival at the ED, adjusted regression analyses

| Total, n= 119 204 aOR* (95% CI) | Office hours† (n=50 417) | Evening shift (n=36 429) | Night shift (n=14 138) | Day shift weekend (n=18 220) |
|---|---|---|---|---|
| Laboratory tests | – | 0.79 | 0.82 | 0.92 |
| | – | (0.76–0.82)‡ | (0.78–0.86)‡ | (0.87–0.96)‡ |
| Imaging | – | 0.93 | 0.64 | 0.99 |
| | – | (0.89–0.96)‡ | (0.60–0.67)‡ | (0.94–1.04) |
| Treatment | – | 1.11 | 1.56 | 1.17 |
| | – | (1.07–1.15)‡ | (1.49–1.63)‡ | (1.12–1.22)‡ |
| Hospital admission | – | 1.06 | 1.32 | 1.05 |
| | – | (1.00–1.11)‡ | (1.24–1.41)‡ | (0.99–1.12) |
| PICU admission | – | 1.15 | 0.92 | 1.19 |
| | – | (0.95–1.42) | (0.71–1.19) | (0.93–1.52) |

*Adjusted for age, gender, urgency, vital signs, presenting problem and hospital.
†Office hours (=day shift week) as reference shift.
‡P value ≤0.05. If not stated: p value>0.05.
aOR, adjusted OR; ED, emergency department; PICU, paediatric intensive care unit.

## DISCUSSION

In this large observational study of 119 204 visits in five EDs in Europe, we demonstrate differences in acuity level and management of children visiting the ED during out-of-office hours compared with office hours.

We found that children visiting during the night were relatively more seriously ill. Second, after correcting for patient characteristics, illness severity and hospital, we observed fewer laboratory and radiological tests while children were more likely to receive treatment at the ED and to be admitted to the hospital especially during the night. No association was found between out-of-office hours visits and PICU admission, after statistically correcting for disease severity.

Overall, the trends in out-of-office hours care were comparable between the different hospitals, with variability remaining.

In a recent study, ED utilisation patterns of paediatric patients throughout the day was analysed.[16] The distribution of visits over the evening and night shifts was quite similar to our results. Also, in line with our results, they observed that children during the night are relatively more ill. So, although the proportion of patients visiting during the night is the smallest, the acuity of this group is the highest, which shows the importance of the same quality of care on a 24-hour-a day basis. In contrast to our results, no association between the shift of arrival and hospital admission was reported. However, this was a single-centre study, which limits generalisability.

Variation in management during out-of-office hours, most pronounced during the night, could not be explained by illness severity. Reducing variation in care could result in lower resource utilisation and costs and could improve the use of time, space and staff.[29] To obtain more standardised care, also during out-of-office hours,

it is essential to understand which factors contribute to the differences we observed.

At first, different levels of staff during the night might influence management. Several studies found differences in resource use[30–34] and admission rates[35] depending on physician background and experience.

Second, the availability of resources during night shifts, such as imaging resources or the availability of specialised radiologists to assess these results during the night, might contribute to the lower use of imaging during the night.[36]

Our results raise the question of whether these higher treatment and admission rates and the lower use of diagnostic imaging during the night might have been unwarranted and contribute to unnecessary costs. Earlier research reported that higher resource utilisation is related to higher costs, but not to lower admission rates.[37–39] One of the hospitals in our study showed no difference in diagnostics during the night, in contrast to the other hospitals, but still showed higher admission rates compared with office hours.

We speculate that our results might also partly be explained by a more pragmatic approach of the patient during the night, resulting in fewer diagnostic tests, more treatment and more hospital admissions. The previous literature did report higher rates of antibiotic prescriptions[17] and recommendations to admit to the hospital,[40] in children visiting the ED during the night. Additionally, parental demand might contribute to these results.[41] Furthermore, in contrast to general hospital admission, the higher rates of PICU admission during the night could be explained by illness severity and case mix. The indications for PICU admissions are more standardised compared with indications for general hospital and therefore less influenced by non-patient factors.

| Table 5 | Hospital characteristics | | | | |
| --- | --- | --- | --- | --- | --- |
| | Erasmus MC-Sophia Children's Hospital, Rotterdam, The Netherlands | Maasstad Hospital, Rotterdam, The Netherlands | St. Mary's Hospital Imperial College Healthcare NHS Trust, London, UK | Hospital Prof. Dr. Fernando Fonseca, Lisbon, Portugal | General Hospital, Vienna, Austria |
| Hospital characteristics | University hospital 60 paediatric beds | Teaching hospital 59 paediatric beds | University hospital 46 paediatric beds | Community hospital 91 paediatric beds | University hospital 134 paediatric beds |
| ED characteristics* | 6500 children/year | 9500 children/year | 27 000 children/year | 60 000 children/year | 22 000 children/year |
| | Urban | Urban | Urban | Mixed urban/rural | Urban |
| | Mixed high and low socioeconomic status | Generally low socioeconomic status | Mixed high and low socioeconomic status | Generally low socioeconomic status | Mixed high and low socioeconomic status |
| No of patients included | 18 590 | 10 583 | 15 556 | 53 175 | 21 300 |
| Inclusion period | 1 January 2012 to 31 December 2014 | 1 May 2014 to 31 October 2015 | 1 July 2014 to 28 February 2015 | 1 March 2014 to 28 February 2015 | 1 January 2014 to 31 December 2014 |
| No of patients during, n (%) | | | | | |
| Office hours | 8915 (48.0) | 4229 (40.0) | 5635 (36.2) | 22 470 (42.3) | 9168 (43.0) |
| Out-of-office hours | 9675 (52.0) | 6354 (60.0) | 9921 (63.8) | 30 705 (57.7) | 12 132 (57.0) |
| Median age, years (IQR) | 4.3 (1.4–9.8) | 5.7 (1.9–11.6) | 3.9 (1.5–8.8) | 4.7 (1.9–9.5) | 3.9 (1.6–8.3) |
| MTS, n (%) | | | | | |
| High urgent | 2427 (13.1) | 1515 (14.3) | 1605 (10.3) | 6222 (11.7) | 1084 (5.0) |
| Urgent | 8744 (47.0) | 5110 (48.3) | 3961 (25.5) | 10 951 (20.6) | 3851 (18.1) |
| Low urgent | 6849 (36.8) | 3857 (36.4) | 9990 (64.2) | 36 002 (67.7) | 15 314 (71.9) |
| Missing | 570 (3.1) | 101 (0.1) | 0 (0.0) | 0 (0.0) | 1051 (4.9) |
| Presenting problem, n (%) | | | | | |
| Medical | 12.206 (65.7) | 5163 (48.8) | 11.520 (74.1) | 43 814 (82.4) | 19 174 (90.0) |
| Trauma | 5814 (31.3) | 5321 (50.3) | 4036 (25.9) | 9361 (17.6) | 1034 (4.9) |
| Missing | 570 (3.1) | 99 (0.9) | 0 (0.0) | 0 (0.0) | 1092 (5.1) |

*All EDs are paediatric only, except for the Maasstad Hospital, which is mixed adult–paediatric. The Erasmus MC-Sophia Children's hospital has a mixed adult–paediatric ED since October 2014 (3 months in total during the inclusion period).
ED, emergency department.

These results point to the need for further research to explain the differences in clinical decisions during the night, by assessing differences in standards of care during out-of-office hours, such as availability of experienced staff and availability of diagnostics and equipment.[14] Moreover, it is important to obtain insight into how these differences in management affect patient outcomes, for example, rehospitalisation, morbidity and 30-day mortality. Implementing guidelines for the management of children with specific conditions could help to reduce variation in management within and between hospitals, which optimise resource allocation and costs. Moreover, these guidelines could allow for better comparisons between the hospitals in further research.

Other studies on the topic of out-of-office hours care, including adult patients at the ED and paediatric patients at the PICU, described worse patient outcomes during the night and evening, represented by higher mortality rates.[1 4 42] We did not use mortality as the main outcome as the mortality in our dataset was very low (16 patients, 0.01%), which is in line with the previous literature.[43 44]

### Strengths and limitations

A major strength of this study is large number of patients from different centres, including both universities as teaching hospitals. This increases the generalisability of the results. Moreover, we adjusted for multiple variables concerning illness severity to analyse the independent role of time of the visit to management.

To appreciate the results, some limitations have to be considered. Hospitals were selected by convenience sampling, which influences the generalisability of the

study. The enrolment period varied between the different hospitals. However, each hospital was followed for at least 8 months and in all hospitals a winter season was included to increase the comparability of the periods. Also, to our knowledge, there were no major changes or differences in for example vaccination status, occurrence of epidemics or other incidents during the total enrolment period (2012–2015).

Second, the analyses were based on routinely collected data. This results in some missing data, which we dealt with using multiple imputation for vital signs. The proportion of missing data for vital signs was comparable with previous studies[45–47] and varied between the hospitals, which can be expected due to patient mix and differences in local policies.

Moreover, it is possible that there are more variables representing illness severity and influencing management, that were not included. These factors can be divided into patient-related and hospital-related factors. Patient-related factors are, for example, comorbidity, socioeconomic status or ethnic background. When looking at hospital-related factors, (experience level of) staff, resource availability and bed availability could possibly contribute to the differences in management. To deal with hospital-related differences, we adjusted our regression analyses for the hospital. Furthermore, we performed the regression analyses for each hospital separately. These stratified analyses demonstrated that overall results were comparable between the hospitals, but some variation remained. However, we did not have the specific information about the aforementioned hospital-level factors for each individual visit to include in the analyses. It would be interesting to analyse the potential effects of these factors in future research.

## CONCLUSION

Children visiting during the night are relatively more seriously ill than those visiting during office hours. This highlights the need to keep improving emergency care on a 24-hour-a-day basis. Furthermore, differences in management are present depending on the time of the ED visit. Especially during the night, diagnostic tests are performed less frequently, while treatment is initiated more often and more children are admitted to the hospital than during office hours. These patterns are not explained by illness severity. Further research should focus on exploring the reasons for these differences and how these differences affect patient outcomes.

**Author affiliations**
¹Department of General Paediatrics, Erasmus MC Sophia Children Hospital, Rotterdam, Zuid-Holland, The Netherlands
²Department of Paediatric Accident and Emergency, Imperial College Healthcare NHS Trust, London, UK
³Department of Paediatrics, Professor Doutor Fernando Fonseca Hospital, Amadora, Lisboa, Portugal
⁴Intensive Care Unit, Professor Doutor Fernando Fonseca Hospital, Amadora, Lisboa, Portugal
⁵Department of Paediatrics, Maasstad Hospital, Rotterdam, Zuid-Holland, The Netherlands
⁶Department of Medical Informatics, Erasmus Medical Center, Rotterdam, Zuid-Holland, The Netherlands
⁷Department of Public Health, Erasmus Medical Center, Rotterdam, Zuid-Holland, The Netherlands
⁸Department of Pediatrics and Adolescent Medicine, Medical University of Vienna, Vienna, Austria

**Contributors** GS and DMB: conceptualisation, data curation, formal analysis, investigation, writing original draft, review and editing. JMZ: data curation, formal analysis, investigation, review and editing. SG-P, CFA, PF, FJS and IKM: data curation, investigation, review and editing. JvdL: conceptualisation, data curation, review and editing. EWS: data curation, review and editing. HAM: conceptualisation, data curation, investigation, project administration, supervision, review and editing.

**Funding** The authors have not declared a specific grant for this research from any funding agency in the public, commercial or not-for-profit sectors.

**Competing interests** None declared.

**Patient and public involvement** Patients and/or the public were not involved in the design, or conduct, or reporting, or dissemination plans of this research.

**Patient consent for publication** Not required.

**Ethics approval** Medical Ethics Committee Erasmus MC: MEC-2013-567 Raad van Bestuur Maasstad Ziekenhuis: Protocol L2013-103 Joint Research Complicance Office Imperial College London and Imperial College Healthcare NHS Trust: 14/WA/1051 Comissão de Ética para a Saúde do Hospital Prof. Dr. Fernando Fonseca, EPE: reunião 06 Dezembro de 2017 Ethik Kommission Medizinische Universität Wien: 1405/2014.

**Provenance and peer review** Not commissioned; externally peer reviewed.

**Data availability statement** Data are available upon reasonable request. Data from this study are available upon request to the corresponding author of the study (h.a.moll@erasmusmc.nl), subject to local rules and regulations.

**ORCID iDs**
Gina Schinkelshoek http://orcid.org/0000-0001-8241-9750
Dorine M Borensztajn http://orcid.org/0000-0002-2437-0757
Joany M Zachariasse http://orcid.org/0000-0002-4093-8509
Henriëtte A Moll http://orcid.org/0000-0001-9304-3322

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
