## [Reviewer comments · BMJ Paediatrics Open]

This paper was submitted to a another journal from Archives of Disease in Childhood but declined for publication following peer review. The authors addressed the reviewers' comments and submitted the revised paper to BMJ Paediatrics Open. The paper was subsequently accepted for publication at BMJ Paediatrics Open.

ARTICLE DETAILS

TITLE (PROVISIONAL)	Management of children visiting the emergency department during out-of-office hours: an observational study
AUTHORS	Schinkelshoek, Gina; Borensztajn, Dorine; Zachariasse, Joany; Maconochie, Ian; Alves, Claudio; Freitas, Paulo; Smit, Frank; van der Lei, Johan; Steyerberg, Ewout; Greber-Platzer, Susanne; moll, Henriette

VERSION 1 – REVIEW

REVIEWER	Reviewer name: Benito, Javier Institution and Country: Hospital de Cruces, Division of Pediatric Emergency Medicine Competing interests: No conflict of interest
REVIEW RETURNED	02-Feb-2020

GENERAL COMMENTS	Thank you for the opportunity to review this manuscript entitled “Management of Children visiting the emergency department during out-of-office hours: an observational study” This is a well-written and interesting study about the characteristics and management of children visiting the emergency department (ED) during out-of-office hours. I think that this study is relevant because although it is known that the profile of children who visit the ED outside office hours, especially at night is different, more information is needed on their special characteristics and possible differences in their management. Knowing these aspects can be important to provide the necessary human and material resources in the different time slots. In this multicenter study, the authors, using a large sample of patients, analyze the different reasons for consultation, their severity and management, depending on the time of day. They find that children who are treated at night show more severity and are hospitalized more frequently. In my opinion this study add knowledge on the subject although there are some aspects that should be commented or clarified. MAJOR CONCERNS There are important differences in the characteristics and management of patients among the five hospitals participating in the study. Around 60% of patients come from two hospitals that see a higher percentage of patients classified as less urgent and with a very low number of trauma consultations. These differences could make the results less generalizable or interfere with their interpretation. The authors show these differences (table 2 and 4 and supplementary table 3) in the last paragraph (inter-hospital
--

	analysis) of the results section but do not mention this possible limitation in the discussion. Taking into account the results, I do not believe that it is possible to assess that children who consult at night are more serious. In fact, children with a low level of urgency consult more frequently during office hours, causing the percentage of more urgent patients to increase at night. On the other hand, the fact that patients were more frequently hospitalized at night was more likely to be due to organizational aspects than to a greater illness severity. Although the multivariable analysis was adjusted for urgency and presenting problem, the higher percentage of patients who consult with certain medical problems at night could explain the differences in management. For example, in children with dyspnea, the emergency physician is more likely to indicate more treatments and fewer diagnostic tests. OTHER COMMENTS Background is adequately reviewed and objectives, methods and results are clearly exposed. One of strengths of this study is that it is a secondary analysis using data from a prospective observational multicenter registry which guarantees the quality of data. Discussion: Pag 13 (line 8, second line of the discussion): I suggest saying instead of: "we demonstrate differences in illness severity....", to say something as "we find differences in the acuity level at triage and management..." . In this way, I think the authors mention this conclusion more adequately in the abstract "Children visiting during the night are relatively more serious...". I make the same suggestion for the phrase on the line 12 in Pag 13. In several paragraphs of the discussion, the authors mention the importance of staffing or the availability of material resources in the different work shifts, over the differences found in the management of patients in the emergency department. However, the lack of information on these aspects in this study is not mentioned as a limitation. Tables and figure are suitable
--	---

REVIEWER	Reviewer name: KOUTOUMANOU, EIRINI Institution and Country: UCL INSTITUTE OF CHILD HEALTH Competing interests: None
REVIEW RETURNED	13-Feb-2020

GENERAL COMMENTS	Thank you for giving me the opportunity to review this interesting report. My main concern about this report is to do with the regression modelling used and why this was not done via hierarchical modelling where hospital would be the main unit. I strongly recommend that the authors repeat their analysis with multilevel modelling and additionally provide answers to the questions below. - Could the authors please add a reference for the TRIAGE project at
--

	the start of the Methods section? Also is there a reference for the Manchester Triage System (MTS) mentioned later in the 'Patient Characteristics' section?  - Could a short comment be added about what led to the specific choice of countries and hospitals? Also, and most importantly, the hospital were followed for very different amounts of time as per Table 4. This needs to be further discussed and the implication involved explained. - In the discussion section, the authors mention that the regression analysis results shown have been stratified for hospital but this needs to be discussed further in the methods and results section. In the methods section, it's mentioned that an interaction term was added for hospital and shift to study inter-hospital differences, but this is not the same as stratification. - How come it was not deemed necessary to collect data on other available socioeconomic/background factors that would normally be routinely collected along with the medical data when exploring those health records? A lot of other factors could potentially have explained the differences seen between shifts, i.e. socioeconomic levels of the are child lives in, ethnical background, etc? - Are there any specific medical incidents/conditions/circumstances that occurred between 2012 and 2015 that should be taken into account? - More information needs to be provided about the multiple imputation of missing data. A flowchart showing missing data per hospital per shift per variable is required in the main part of the paper, but additional details regarding the MI model could possibly be included in the supplementary material section (how many imputations, what software used, what variables involved in the imputation, etc). Minor comment: use commas instead of dots to indicate thousands
--	---

VERSION 1 – AUTHOR RESPONSE

Reviewer: 1

- MAJOR CONCERNS

There are important differences in the characteristics and management of patients among the five hospitals participating in the study. Around 60% of patients come from two hospitals that see a higher percentage of patients classified as less urgent and with a very low number of trauma consultations. These differences could make the results less generalizable or interfere with their interpretation. The authors show these differences (table 2 and 4 and supplementary table 3) in the last paragraph (inter-hospital analysis) of the results section but do not mention this possible limitation in the discussion.

Response: We agree that there are possibly more differences at hospital-level that may influence management, that we could not include in our analyses (for example staff level). To take these differences between hospitals into account, we adjusted the multivariable regression analyses for 'hospital' in the original manuscript. Moreover, the regression analyses were adjusted for age, gender, abnormal vital signs, urgency level and for presenting problem, Hereby we aimed to study the independent role of shift as accurate as possible. (Methods 'Statistical analyses' section, page 8.

Results 'Management' section, page 11. Table 4, page 24). To provide further insight into the differences between the hospitals, we performed stratified analyses, by performing the regression analyses for each hospital separately. (Supplementary table 4, page 3)

To emphasize that it is possible that there are more differences at hospital-level that were not included, we have revised the Discussion 'Strengths and Limitations' section (page 14-15):

Secondly, it is possible that there are more variables representing illness severity and influencing management, that were not included. These factors can be divided into patient- and hospital-related factors. Patient-related factors are e.g. comorbidity, socio-economic status or ethnic background. When looking at hospital-related factors, (experience level of) staff, resource availability and bed availability could possibly contribute to the differences in management. To deal with hospital-related differences, we adjusted our regression analyses for hospital. Furthermore, we performed the regression analyses for each hospital separately. These stratified analyses demonstrated that overall results were comparable between the hospitals, but some variation still remained. However, we did not have the specific information about the aforementioned factors for each individual visit, to include in the analyses. It would be interesting to analyse the potential effects of these factors in future research.

-Taking into account the results, I do not believe that it is possible to assess that children who consult at night are more serious. In fact, children with a low level of urgency consult more frequently during office hours, causing the percentage of more urgent patients to increase at night. On the other hand, the fact that patients were more frequently hospitalized at night was more likely to be due to organizational aspects than to a greater illness severity. Although the multivariable analysis was adjusted for urgency and presenting problem, the higher percentage of patients who consult with certain medical problems at night could explain the differences in management. For example, in children with dyspnoea, the emergency physician is more likely to indicate more treatments and fewer diagnostic tests.

Response: We agree that our results suggest that children during the night are relatively more ill than during office hours. The results of the adjusted multivariable regression analyses suggest that the differences in management could not be explained by illness severity. This is mentioned in the Discussion section 'Conclusion' (page 15):

Children visiting during the night are relatively more seriously ill than those visiting during office hours. This highlights the need for optimal emergency care on a 24 hours a day basis. Furthermore, differences in management are present depending on the time of ED visit. Especially during the night, diagnostic tests are performed less frequently, whilst treatment is initiated more often and more children are admitted to the hospital than during office hours. These patterns are not explained by illness severity.

Thank you for the suggestion to take the medical problem into account. We have incorporated the presenting problems into our analyses. Instead of adjusting for 'medical' or 'trauma' in the regression analyses, we redid our analyses and adjusted for subgroups of presenting problems.

Furthermore, to perform these revised analyses we regrouped the different presenting problems in 9 (larger) subgroups. (see also Methods section 'Patient characteristics, page 7):

- Respiratory
- Ear, nose and throat (ENT) problems
- Gastro-intestinal problems
- Neurological or psychiatric problems

- General malaise
- Uro- or gynaecological problems
- Dermatologic
- Trauma or muscular
- Others

After adjusting the analyses for presenting problem, age, gender, urgency level, abnormal vital signs and hospital, we still found that laboratory and imaging diagnostic were significantly less likely to be performed and treatment was significantly more likely to be given during out-of-office hours, especially the night. Also, admission to the hospital was significantly more likely during the night. (Table 4 The association between management (lab, imaging, treatment, hospital and PICUa admission) and time of arrival at the ED, adjusted regression analyses, page 24)

This adaption of the subgroups of presenting problem is also implemented in: Results 'Patient characteristics' section. (page 10 main document) Figure 1. Presenting Problems. (separate file/image) Table 5 Hospital Characteristics 'Presenting Problem'. (page 24 main document) Supplementary table 3. (page 2 Supplementary file)

For additional information concerning these subgroups, we added supplementary table 2 (page 1 Supplementary file).

- OTHER COMMENTS

Discussion:

Pag 13 (line 8, second line of the discussion): I suggest saying instead of: "we demonstrate differences in illness severity...", to say something as "we find differences in the acuity level at triage and management..." . In this way, I think the authors mention this conclusion more adequately in the abstract "Children visiting during the night are relatively more serious...". I make the same suggestion for the phrase on the line 12 in Pag 13.

Response: Thank you for this suggestion, we have changed the sentences you have pointed out (Discussion, page 12):

We demonstrate differences in acuity level and management of children visiting the ED during out-of-office hours compared to office hours.

We found that children visiting during the night were relatively more seriously ill.

- In several paragraphs of the discussion, the authors mention the importance of staffing or the availability of material resources in the different work shifts, over the differences found in the management of patients in the emergency department. However, the lack of information on these aspects in this study is not mentioned as a limitation.

Response: We agree that this aspect was not discussed clearly enough in the limitation section. To point out this aspect, we have now adapted the discussion 'Strengths and Limitations' section (page 15): When looking at hospital-related factors, (experience level of) staff, resource availability and bed availability could possibly contribute to the differences in management. To deal with hospital-related differences, we adjusted our regression analyses for hospital. Furthermore, we performed the regression analyses for each hospital separately. These stratified analyses demonstrated that overall results were comparable between the hospitals, but some variation still remained. However, we did not have the specific information about the aforementioned factors for each individual visit to include

in the analyses. It would be interesting to analyse the potential effects of these factors in future research.

Reviewer: 2

Comments to the Author

- Thank you for giving me the opportunity to review this interesting report. My main concern about this report is to do with the regression modelling used and why this was not done via hierarchical modelling where hospital would be the main unit. I strongly recommend that the authors repeat their analysis with multilevel modelling and additionally provide answers to the questions below.

Response: Thank you for raising this point. We agree that a hierarchical model would seem appropriate, because of the clustered nature of the data and the potential effects of these clusters on the results. Our main motivation not to use the multilevel model was that our dataset only includes five hospitals and thereby we expected that the reliability of the estimates of the inter-hospital differences would not be fulfilling. For this reason, we choose to use the multivariable regression analyses with hospital as fixed effect, to account for possible bias by inter-hospital differences. (Table 4, page 23) Additionally, we choose to perform stratified analyses, by conducting the regression analyses for each hospital separately, to present the inter-hospital differences as accurate as possible. (Supplementary table 4, page 3)

Based on your suggestion, we did perform the multilevel analyses with a random intercept for hospital. For all outcome variables (laboratory diagnostics, imaging diagnostics, treatment and hospital admission), the intercept-only model resulted in a non-significant estimate of the random effect.

When fitting an adjusted model, including shift, age, gender, urgency level, presenting problem and vital signs as fixed effect and a random intercept for hospital, the results (Odds ratios and Confidence Interval) were similar to the results obtained from our regression analyses.

- Could the authors please add a reference for the TRIAGE project at the start of the Methods section? Also is there a reference for the Manchester Triage System (MTS) mentioned later in the 'Patient Characteristics' section?

Response: A reference for another study using the TRIAGE dataset is added now. (Methods section 'Study design, setting, patients' page 6):

Zachariasse JM, Nieboer D, Maconochie IK, Smit FJ, Alves CF, Greber-Platzer S, Tsolia MN Steyerberg EW, Avillach P, van der Lei J, Moll HA. Development and validation of a Paediatric Early Warning Score for use in the emergency department: a multicentre study. Accepted for publication in Lancet Child Adolesc Health. 2020.

The reference for the Manchester Triage System (MTS) was mentioned in the Methods section 'Patient Characteristics' (page 7). To further meet your point, we have added Supplementary table 2 in the Supplementary File (page 1). This table provides information about which presenting problems of the MTS Flowchart (for example 'Ear problems', 'Sore throat' and 'Facial problems') are included in the subgroups of presenting problems (for example 'Ear, Nose, Throat').

- Could a short comment be added about what led to the specific choice of countries and hospitals? Also, and most importantly, the hospital were followed for very different amounts of time as per Table 4. This needs to be further discussed and the implication involved explained.

The choice of countries and hospitals was based on convenience sampling. As requested, a short comment is added in the Methods section 'Study design, setting, patients' (page 6): 'The five

participating hospitals, selected by convenience sampling, were: ...' We have mentioned the different sampling periods in the discussion section 'Strengths and Limitations' (page 14) now:

To appreciate the results, some limitations have to be considered. The enrolment period varied between the different hospitals. However, each hospital was followed for at least 8 months and in all hospitals a winter season was included to increase the comparability of the periods. Also, to our knowledge, there were no major changes or differences in for example vaccination status, occurrence of epidemics or other incidents during the total enrolment period (2012-2015).

The aim of the study was to follow all the hospitals for a minimum of one year, which was accomplished in 4 of the 5 participating hospitals. In one hospital (United Kingdom), a year was not achievable, because data extraction was not possible, due to a change in the patient record system.

- In the discussion section, the authors mention that the regression analysis results shown have been stratified for hospital but this needs to be discussed further in the methods and results section. In the methods section, it's mentioned that an interaction term was added for hospital and shift to study inter-hospital differences, but this is not the same as stratification.

Response: Thank you for pointing this out. To report the analyses more clearly, we have adapted the methods section 'Statistical analysis' (page 8-9). We have now stated:

To study the independent role of shift of arrival, the regression analyses were adjusted for age, gender, MTS urgency, abnormal vital signs, presenting problem and hospital. We performed stratified analyses, by conducting the adjusted regression analyses for each hospital separately, to provide insight into inter-hospital differences.

To also clarify this point in the results section 'Inter-hospital analyses' (page 11), we have now stated:

The multivariable regression analyses, adjusted for age, gender, urgency, abnormal vital signs and presenting problem, were performed for each hospital separately in stratified analyses. (supplementary table 4) Overall patterns in management during out-of-office hours were comparable between the different hospitals, with some variation remaining.

- How come it was not deemed necessary to collect data on other available socioeconomic/background factors that would normally be routinely collected along with the medical data when exploring those health records? A lot of other factors could potentially have explained the differences seen between shifts, i.e. socioeconomic levels of the area child lives in, ethnical background, etc?

Response: We definitely agree that it would be interesting to explore socioeconomic factors. In our study, we have used a database based on routinely collected data. Socioeconomic status or ethnical background of the individual patients were not included in this data. Therefore, we have now incorporated this limitation in the discussion section 'Strengths and Limitations' (page 14): Secondly, it is possible that there are more variables representing illness severity and influencing management, that were not included. These factors can be divided into patient- and hospital-related factors. Patient-related factors are for example comorbidity, socio-economic status or ethnic background.

Moreover, we have added general information about the characteristics of the catchment area (socio-economic status) of the different hospitals in Table 5. Hospital Characteristics (page 24).

- Are there any specific medical incidents/conditions/circumstances that occurred between 2012 and 2015 that should be taken into account?

Response: To our best knowledge, there are no medical circumstances of which we expect that it would considerably influence the results or the generalizability of the results. For example, we checked if there were any major changes in vaccination status in different countries during 2012-2015. Also, the enrolment period did not coincide with the H1N1 epidemic (2009-2010) or the global financial crisis (2007-2008). Moreover, no unusual conditions or incidents were reported by the leads of the included Emergency Departments.

- More information needs to be provided about the multiple imputation of missing data. A flowchart showing missing data per hospital per shift per variable is required in the main part of the paper, but additional details regarding the MI model could possibly be included in the supplementary material section (how many imputations, what software used, what variables involved in the imputation, etc).

Response: Thank you for highlighting this. We agree that additional information about the multiple imputation is needed. We have added 'Table 1. Missing vital signs and imputation model' to the main document (page 21). In this table we are showing the missing data per variable, including inter-hospital ranges, and information about the multiple imputation model. We are aware that the suggestion you made was to show the missing data per variable per shift per hospital. However, after performing this, we believed that adding this large amount of data to the main document, would interfere with the readability of the manuscript. Therefore, we included graphs with the missing data per variable per shift per hospital, to the supplementary file. (supplementary figure 1, page 4) If a different presentation of these results is preferred, we are open to suggestions.

- Minor comment: use commas instead of dots to indicate thousands

Response: The dots indicating thousands are changed to commas.

As a result of incorporating all the feedback, the word count of our manuscript increased by 141 words and thereby exceeded the maximum amount of 2500 words. We would be pleased to hear if we may accept this count at your discretion.

We look forward to hearing from you soon and to respond to any further questions or comments you may have.

VERSION 2 – REVIEW

REVIEWER	Reviewer name: Javier Benito Institution and Country: Cruces University Hospital Basque-Country - Spain Competing interests: No
REVIEW RETURNED	18-May-2020
GENERAL COMMENTS	Authors have addressed all my concerns and suggestions. Nice work here!
REVIEWER	Reviewer name: Eirini Koutoumanou Institution and Country: UCL, UK Competing interests: none
REVIEW RETURNED	27-May-2020
GENERAL COMMENTS	Thank you for the resubmission and careful consideration of all reviewers' comments. Two main concerns remain from my end: one about the regression analysis results and one about the missing data.

Regression analysis:

The results of the stratified regression analysis clearly show, as discussed by the authors too, that there is some variability between hospitals (Table 5 and Supplementary Table 4).

Just one example, the odds of hospital admissions at St Mary's were 5% lower during night shift compared to day shifts (albeit not significant, but CI fairly wide, 0.8 to 1.19).

Also, for some cases the most noticeable difference came between weekend day and weekday day shifts (or other pairwise comparison beyond day and night). It might be worth adding a comment about these occurrences.

Therefore, I recommend that the following statement is slightly altered to reflect this variability: "Overall, the trends in out-of-office hours care were comparable between the different hospitals."

The authors also mention that they tried multilevel regression analysis with a random intercept per hospital but was not significant. Did they also fit a random slope for each hospital?

Missing data for vital signs:

I strongly recommend that the term 'missing vital signs' is replaced by 'missing data for vital signs' or even 'missing data for the vital signs variable' for additional clarity. Also:

- Could the authors please make a brief comment about the pattern of missingness? Was it as expected between hospitals (as the missing rate for some of the signs is very high for some hospitals compared to others, Table 1 widest range 5.9-58.2)?

- What are the authors views on the Missing at Random and other missingness assumptions?

- Did they perform sensitivity analysis firstly for a larger number of imputations (currently reported equal to 25) and also against results from listwise deletion? Both of these are necessary when reporting results from multiple imputations. [see Box 2 of Sterne, Jonathan AC, et al. "Multiple imputation for missing data in epidemiological and clinical research: potential and pitfalls." *Bmj* 338 (2009): b2393.]

Finally, both in the abstract and conclusion section of the paper, the authors make reference to "...optimal emergency care on a 24 hours a day basis". I do not believe that this study has shown evidence towards a suboptimal level of care – that was not the aim anyway (as far as I've understood it), so I'd recommend rephrasing.

Minor comments:

- The number of patients in the abstract has not been changed to a comma from a dot.

- Supplementary material table 2 is referenced before supplementary table 1 in the main text.

- Add a small comment about the convenience sampling in the limitations sections as it affects generalisability of the results.

- Table 5 is very informative and it would be even more interesting to see the Age split in 4 rows for each hospital per shift group (ideally I'd like the MTS and presenting symptoms split like that too) – maybe this could be an extra supplementary table as I think it'd extremely informative.

VERSION 2 – AUTHOR RESPONSE

We are very thankful that our manuscript is recommended for publication. Moreover, we greatly appreciate the time and effort that is dedicated again to reviewing our manuscript carefully. We have incorporated the feedback into our revised manuscript, the changes are highlighted within the attached manuscript. Below you will find a point-by-point response to the reviewers' comments and concerns.

Regression analysis:

The results of the stratified regression analysis clearly show, as discussed by the authors too, that there is some variability between hospitals (Table 5 and Supplementary Table 4).

Just one example, the odds of hospital admissions at St Mary's were 5% lower during night shift compared to day shifts (albeit not significant, but CI fairly wide, 0.8 to 1.19).

Also, for some cases the most noticeable difference came between weekend day and weekday day shifts (or other pairwise comparison beyond day and night). It might be worth adding a comment about these occurrences.

Therefore, I recommend that the following statement is slightly altered to reflect this variability: "Overall, the trends in out-of-office hours care were comparable between the different hospitals."

We agree with your point here. According to your suggestion, we have reported these differences as followed:

Overall patterns in management during out-of-office hours were comparable between the different hospitals, with some variation remaining. For example, at two hospitals no significant difference was found in hospital admissions during the night. At another hospital imaging was requested more frequently during evening and weekend shifts. (supplementary table 4). (Results – Inter-hospital analyses, page 11)

In the conclusion we have altered the statement:

Overall, the trends in out-of-office hours care were comparable between the different hospitals, with some variability remaining. (Discussion, page 12)

Also, in the Abstract - Results (page 9) we have stated:

Patterns in management during out-of-office hours were comparable between the different hospitals, with variability remaining.

The authors also mention that the tried multilevel regression analysis with a random intercept per hospitals but was not significant. Did they also fit a random slope for each hospital?

We did not perform a multilevel model with a random slope, as we assumed that, in this study, we did not have a sufficient number of clusters to obtain accurate estimates of separate slopes for each hospital. However, we highly appreciate your feedback and suggestions about the multilevel model and will definitely be able to use this in further research.

Missing data for vital signs:

I strongly recommend that the term 'missing vital signs' is replaced by 'missing data for vital signs' or even 'missing data for the vital signs variable' for additional clarity.

Thank you for this suggestion. As recommended, we have adjusted the term 'missing vital signs' to 'missing data for vital signs'. (Methods – Statistical Analysis, page 9. Table 1, page 21. Supplementary figure 1, page 7).

Could the authors please make a brief comment about the pattern of missingness? Was it as expected between hospitals (as the missing rate for some of the signs is very high for some hospitals compared to others, Table 1 widest range 5.9-58.2)?

Since we used routinely collected data, we expected to have missing values. The proportion of these missing data was comparable with other studies (1-3). As mentioned, we found variability in the proportion of missingness between the hospitals, which can be expected as each hospital has differences in local policies and compliance to these policies. Also, there were differences in the characteristics of the patient population between the hospitals. As these analyses were descriptive, there was not yet adjusted for patient mix. We expected different patients factors to influence the pattern of missingness, which will be discussed in the next comment.

We have now stated in the Discussion section – Strengths and Limitations (page 14-15): Secondly, the analyses were based on routinely collected data. This results in some missing data, which we dealt with using multiple imputation for vital signs. The proportion of missing data for vital signs was comparable with previous studies (1, 2, 4) and varied between the hospitals, which can be expected due to patient mix and differences in local policies.

To further emphasize this point, we have added information to the supplementary appendix.

Supplement 3. Missing data for vital signs (page 4-5, supplementary appendix):

A variety in the proportion of missingness was observed between the different hospitals, which is likely due to differences in characteristics of the patient populations and in local policies and compliance to these policies. (Supplementary figure 1)

What are the authors views on the Missing at Random and other missingness assumptions?

We assumed the missing data for vital signs to be missing at random, as we expected them to be conditional on other variables that were included in our database. This assumption was supported by exploratory analyses, where we found strong associations between both patient and setting factors. In the multiple imputation model we included general patient characteristics, date and time of visit, triage items, vital signs, diagnostics, therapy and disposition. So both the outcome variables, as the independent variables that we have used in our regression analyses, were included in the imputation model. To present this information more clearly, we have added supplementary table 3. Multiple imputation model. (supplementary appendix, page 4).

Also we have provided the following additional information about the imputation model (supplement 3, page 4 supplementary appendix):

The missing data for the vital signs variables (heart rate 42.7%, respiratory rate 52.4%, oxygen saturation 44.9%, temperature 22.5%) were assumed to be missing at random. We expected that the pattern of missingness could be explained by different patient and setting-related factors, that were included in the dataset. For example, based on clinical experience, measuring vital signs was considered less useful or relevant in low urgency cases or in patients with specific presenting problems, such as simple fractures. Also in high urgency cases data of vital signs was missing, which was not expected. This could be partly explained by data of vital signs that were not documented in the patient record, instead of vital signs that were not measured. The proportion of missing data for vital signs was comparable with previous studies. (1-3) A variety in the proportion of missingness was observed between the different hospitals, which is likely due to differences in characteristics of the patient populations and in local policies and compliance to these policies. (Supplementary figure 1) In exploratory analyses, strong associations were found between the proportion of missing data and several patient and hospital-related factors (type of presenting complaint, triage urgency and disposition after the ED visit), thereby supporting the assumption that the missing data for vital signs were missing at random.

The missing data were imputed using a multiple imputation model including general patient characteristics, date and time of visit, triage items, vital signs, diagnostics, therapy and disposition.

(Supplementary table 3)

Imputation was performed by using the MICE imputation package in R, version 2.15.2. The imputation process resulted in twenty-five datasets on which statistical analyses were performed and pooled for a final result.

Moreover, we have added to the Methods section – Statistical Analyses (page 9):

Missing data for the vital signs variable were imputed using a multiple imputation model. Missing data for vital signs were assumed to be conditional on other variables included in the database, i.e. missing at random.

Did they perform sensitivity analysis firstly for a larger number of imputations (currently reported equal to 25) and also against results from listwise deletion? Both of these are necessary when reporting results from multiple imputations. [see Box 2 of Sterne, Jonathan AC, et al. "Multiple imputation for missing data in epidemiological and clinical research: potential and pitfalls." *Bmj* 338 (2009): b2393.]

Thank you for addressing the issue of performing sensitivity analyses. Our statistical analyses were performed on a total of 25 imputations. To our best knowledge, this should be a sufficient number to reduce sampling variability from the imputation process. Also, the paper by Sterne et al. already considers 20 imputations a large number.

According to your suggestion, we have conducted additional analyses in a database consisting only of complete cases for vital signs. The results are presented in the following table: Supplementary table 4 The association between management (lab, imaging, treatment, hospital and PICUa admission) and time of arrival at the ED, adjusted regression analyses. Complete cases (vital signs). (page 6, Supplementary appendix)

The complete case analyses and the results are also described in the supplementary file (Supplement 4. Complete case analyses, page 6):

For the complete case analyses, we used a database consisting only of cases in which all vital signs (heart rate, respiratory rate, oxygen saturation, temperature) were measured and documented (total n=51,664). We performed the same regression analyses as with the imputation database, adjusted for age, gender, urgency, vital signs, presenting problem and hospital.

We found comparable results of the association between management and time of arrival at the ED, when comparing the results of the imputation dataset to the complete case dataset (Supplementary table 4). In the complete case analyses, imaging was less likely to be requested and treatment was more likely to be given during all out-of-office hours. During the night patients were significantly more likely to be admitted to the hospital. In complete case analyses, only the slightly lower number of laboratory diagnostics and slightly higher number of hospital admissions was not seen in weekend day shifts, but was confirmed in evening shifts. More chance variation in complete case analyses is likely to contribute to these differences. (5, 6)

We have added a comment about these analyses to the main document, Methods section – Statistical analysis (page 9): Sensitivity analyses were performed by analysing a database restricted to complete cases. (supplement 4).

Also the results are mentioned briefly in the main document, Results section – Management (page 11):

When analysing a database consisting only of complete cases for vital signs, overall results were comparable to the analyses in the imputation database. (supplement 4)

Finally, both in the abstract and conclusion section of the paper, the authors make reference to "...optimal emergency care on a 24 hours a day basis". I do not believe that this study has shown evidence towards a suboptimal level of care – that was not the aim anyway (as far as I've understood it), so I'd recommend rephrasing.

We agree with your point here. Although we do believe that reducing variation in care could result in lower resource utilization and costs, and could improve the use of time, space and staff, our study could not conclude that there is direct evidence for a suboptimal level of care. In response to this comment, we have rephrased this statement in the Abstract (page 3):

Children visiting during the night are relatively more seriously ill, highlighting the need to keep improving emergency care on a 24 hours a day basis.

In the Discussion section – Conclusion (page 15) we have stated:

Children visiting during the night are relatively more seriously ill than those visiting during office hours. This highlights the need to keep improving emergency care on a 24 hours a day basis.

The number of patients in the abstract has not been changed to a comma from a dot.

Thank you, this is changed in the abstract now.

Supplementary material table 2 is referenced before supplementary table 1 in the main text.

The numbers of the tables are switched now. To further improve the clarity and readability of the supplementary appendix, we have added an index page and grouped the supplementary files by subject.

Add a small comment about the convenience sampling in the limitations sections as it affects generalisability of the results.

According to your comment, we have now added:

To appreciate the results, some limitations have to be considered. Hospitals were selected by convenience sampling, which influences the generalizability of the study. Results section – Strengths and Limitations (page 14).

Table 5 is very informative and it would be even more interesting to see the Age split in 4 rows for each hospital per shift group (ideally I'd like the MTS and presenting symptoms split like that too) – maybe this could be an extra supplementary table as I think it'd extremely informative.

We have welcomed your suggestion and included five additional tables to the supplementary file, including patient characteristics (age, gender, MTS urgency and presenting problem) for each hospital separately. (Supplement 5, supplementary table 5.1-5.5, page 7-9)

We hope that the responses and changes are satisfactory and that the manuscript is now suitable for publication.
